# Snakebite envenomations and access to treatment in communities of two indigenous areas of the Western Brazilian Amazon: A cross-sectional study

Altair Seabra de Farias[1,2], Manoel Rodrigues Gomes Filho[3], Macio da Costa Arévalo[3], Joseir Saturnino Cristino[1,2], Franciane Ribeiro Farias[4], André Sachett[1,2], Alexandre Vilhena Silva-Neto[1,2], Fabíola Guimarães de Carvalho[1], Sediel Andrade Ambrosio[5], Erica da Silva Carvalho[1,2], Marcus Lacerda[1,2,6], Felipe Murta[1,2], Vinícius Azevedo Machado[1], Fan Hui Wen[7], Wuelton Monteiro◉[1,2]*, Jacqueline Sachett[1,2]

**1** School of Health Sciences, Universidade do Estado do Amazonas, Manaus, Brazil, **2** Research Department, Fundação de Medicina Tropical Dr. Heitor Vieira Dourado, Manaus, Brazil, **3** Distrito Sanitário Especial Indígena Alto Rio Solimões, Secretaria Especial de Saúde Indígena, Tabatinga, Brazil, **4** Centro de Estudos Superiores de Tabatinga, Universidade do Estado do Amazonas, Tabatinga, Brazil, **5** Faculdade de Medicina, Universidade Federal do Amazonas, Manaus, Brazil, **6** Instituto Leônidas & Maria Deane, Fiocruz, Manaus, Brazil, **7** Instituto Butantan, São Paulo, Brazil

* wueltonmm@gmail.com

**Data Availability Statement:** Data underlying the findings are fully available in the manuscript supplementary files.

## Abstract

### Background

The indigenous populations of Brazil present poor health indicators and a disproportionate prevalence and case-fatality rate of neglected tropical diseases, including snakebite envenomations (SBEs). This study aims to estimate access to medical care for SBEs and analyze the barriers that prevent victims from accessing healthcare in indigenous communities in two health districts located in the Western Brazilian Amazon.

### Methodology/Principal findings

This cross-sectional study used semi-structured interviews to collect data from individuals who experienced SBEs in the Upper Rio Solimões and Upper Rio Negro indigenous health districts. Of the 187 participants, 164 (87.7%) reported that they had access to healthcare and received assistance in a hospital in the urban area of the municipalities. Frequency was 95.4% in the Upper Rio Solimões SIHD, and 69.6% in the Upper Rio Negro SIHD (P<0.0001). The study found that the availability of indigenous medicine as the only choice in the village was the main reason for not accessing healthcare (75%), followed by a lack of financial resources and means of transportation (28.1%). Four deaths were reported from SBEs, resulting in a case-fatality rate of 2.1%.

### Conclusions/Significance

In the study areas, there are records of SBE patients who did not receive medical attention. Availability of pre-hospital emergency transport using motorboats, a greater number of

**Funding:** J.S. and W.M. are funded by Conselho Nacional de Desenvolvimento Científico e Tecnológico (CNPq productivity scholarships). W. M. and J.S. were funded by Fundação de Amparo à Pesquisa do Estado do Amazonas (PRÓ-ESTADO, call 011/2021 - PCGP/FAPEAM, call 010/2021 - ÁREAS PRIORITÁRIAS, call 023/2022 - INICIATIVA AMAZÔNIA +10) and by the Ministry of Health, Brazil (proposal No. 733781/19-035). A.S.F. and F. M. are funded by Fiocruz (Inova scholarships). A.S. F., F.M. and M.L. and F.M. were funded via Programa Inova Fiocruz and VPAAPS/Fiocruz, project "Contribuição para o desenvolvimento de estratégias para o fortalecimento do SasiSUS, considerando as vulnerabilidades emergentes e reemergentes em saúde". The funders had no role in study design, data collection and analysis, decision to publish, or preparation of the manuscript.

**Competing interests:** The authors have declared that no competing interests exist.

hospitals and better navigability of the Solimões River and its tributaries would make access easier for indigenous people living in the region of the Upper Solimões River. The implementation of cross-cultural hospital care needs to be considered in order to reduce the resistance of indigenous populations in relation to seeking treatment for SBEs.

## Author summary

Interdisciplinary research on snakebites and the engagement of indigenous and riverine populations of the Amazon are essential for the formulation of innovative and tailored strategies towards achieving the global target of halving the number of deaths and disabilities due to snakebite envenomations by 2030 in Brazil. These populations are key to reducing these poor outcomes since they are disproportionately affected by snakebites. In this work, we evaluated access to medical care in SBE patients and analyzed barriers that prevent victims from accessing healthcare in indigenous villages in two health districts in the Brazilian Amazon. Snakebites were reported in all age groups, including a high rate among children, which demonstrates their great contact with venomous snakes in indigenous villages. A proportion of 12.3% of individuals with a lifetime history of snakebites did not receive hospital treatment, ranging from 4.6% in the Upper Rio Solimões health district to 30.4% in the Upper Rio Negro health district. Availability of pre-hospital emergency transport using motorboats, a greater number of hospitals and better navigability of the Solimões River and its tributaries would make access easier for indigenous people living in this district. We discuss the role that the implementation of cross-cultural hospital care, which considers specific indigenous needs for accommodation and food, may have in reducing the resistance of indigenous populations to seek western treatment for snakebites.

## Introduction

In 2018, the World Health Organization (WHO) launched a strategy to reduce mortality and disability rates caused by snakebite envenomations (SBEs) by half by 2030 [1]. This strategy includes ensuring safe, accessible and effective treatment and empowering communities to be proactive in preventing bites, improving access to treatment, and strengthening local health systems to achieve better outcomes for patients. In Brazil, a plan to achieve self-sufficiency in manufacturing snakebite antivenoms by state laboratories was developed in the 1980s, and this was coupled with a national epidemiological surveillance system that is organized to distribute these immunobiological drugs to hospitals in almost 3,000 municipalities, where free of charge antivenom treatment is provided [2]. Despite all the advances in this system, antivenoms are not uniformly available or accessible across the Brazilian territories, principally in the remote areas in the Brazilian Amazon [2,3]. Lack of timely access to antivenoms is a major determinant of the disproportionately high morbidity and mortality rates associated with SBEs in the Brazilian Amazon when compared to the rest of the country [2,3].

In the Brazilian Amazon, the therapeutic itineraries of SBE patients are affected by the low acceptance and poor infrastructure of local healthcare facilities, which are a result of the ineffective response to health problems in the municipalities [4]. The current plans to increase antivenom treatment coverage in remote areas of the Amazon require infrastructure investments in safe storage options for antivenoms and increased access to fully equipped treatment

facilities. Decentralization of antivenom treatment must be coupled with professional training to strengthen the local health systems [3]. Barriers and facilitators for the implementation processes must include the analysis of organizational priorities, resources and capabilities, and be aligned with environmental and cultural components of the healthcare system, including staff understanding, commitment and attitudes. Furthermore, in SBE-endemic countries, healthcare workers lack sufficient knowledge to be able to manage envenomed patients and feel insecure when treating SBEs, which is caused by limited training of these professionals during their medical education [5,6].

The Brazilian indigenous population, according to the results of the last national census in 2010, was 896,917 indigenous people, of which 572,083 lived in rural areas and 324,834 lived in urban areas. From the total, 305,873 (37.4%) lived in the northern region of the country; in this region, the state with the largest indigenous population is the state of Amazonas, with 55% of the total [7]. In Brazil, poor health indicators are reported in indigenous populations, with a disproportionate prevalence and case-fatality of neglected tropical diseases [8]. Importantly, there is a substantial, but so far unmeasured. disease burden from tuberculosis [9], malaria [10], leishmaniasis [11], soil-transmitted helminthiases [12], Chagas disease [13], trachoma [14], leprosy [15], and SBEs [3]. However, there is still a need for transdisciplinary solutions to the problem of neglected diseases in the Amazonian indigenous communities [16].

This study aimed to estimate the proportion of access to healthcare in SBE cases and analyze barriers that prevent victims from obtaining healthcare in indigenous communities located in two indigenous health districts, in the Western Brazilian Amazon.

## Methods

### Ethics statement

This study involved collection of data from indigenous populations, and the consent was obtained from indigenous leaders from each village. After this consent was obtained, the study protocol was submitted to the Health Research Coordination of the National Council for Scientific and Technological Development (COSAU/CNPq) and to the National Indigenous Foundation (FUNAI). Subsequently, with the approvals from COSAU/CNPq and FUNAI, the protocol was submitted to and approved by the Amazonas State University Ethical Board and the National Research Ethics Commission (approval number 4,993,083/2021). FUNAI issued the authorization for entry into indigenous areas under number 3/AAEP/PRES/2021. However, due to the COVID-19 pandemic, researchers' entries into indigenous lands were suspended. Therefore, data collection was carried out by three nurses providing health care in the study area, after training to apply the research instrument. To ensure understanding of the study, the interviews were always carried out accompanied by a native speaker of the participant's language. All participants signed a consent form after full reading of the study's objectives and procedures. Children and adolescents signed an assent form and their parents or legal guardians signed a consent form agreeing to the inclusion of the minor in the study.

### Study area

This study was carried out in the area covered by two special indigenous health districts (SIHDs; Upper Rio Negro and Upper Rio Solimões) in the state of Amazonas, in the Western Brazilian Amazon. These areas are demarcated by the federal government, and border Peru and Colombia (Fig 1). This region is the one with the greatest burden of SBEs in the state of Amazonas, in which incidence rates reach over 150 SBEs per 100,000 inhabitants/year [17], and the case-fatality rate is 40 deaths from SBEs/1,000 cases [18].

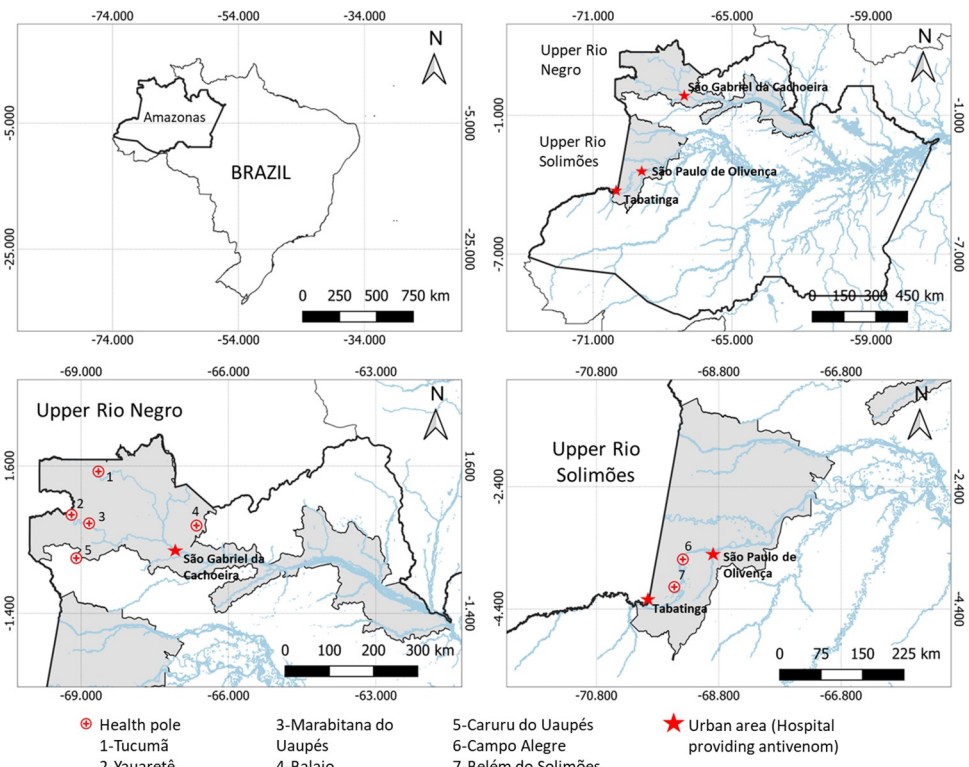

**Fig 1. Study area.** A: State of Amazonas, in Brazil; B: Area of the Special Indigenous Health Districts (SIHD; Upper Rio Negro and Upper Rio Solimões), in the Western Brazilian Amazonia; C: Upper Rio Negro SIHD area, with the urban area of São Gabriel da Cachoeira and five health centers; D: Area covered by the Upper Rio Solimões SIHD, with the urban areas of Tabatinga and São Paulo de Olivença and two health centers. The geographical coordinates of each health center were obtained during the visits using GPS (Garmin GPSMAP 64x). The base used to create the map was obtained from the Brazilian Institute of Geography and Statistics, which is freely accessible for creative use in shapefile format, in accordance with the Access to Information Law (12,527/2011) (https://www.ibge.gov.br/geociencias/downloads-geociencias.html?caminho=cartas_e_mapas/bases_cartograficas_continuas/bc250/versao2021/).

**Upper Rio Negro SIHD.**    In its area of coverage, it has 25 health centers, 19 of which are in São Gabriel da Cachoeira, that are distributed in 747 villages and towns, and 32,720 indigenous people in an area of 138,020.94 km$^2$. The upper Negro River region is inhabited by 23 indigenous peoples who speak the languages of the Eastern Tukano, Aruak and Maku families. Within the Eastern Tukano linguistic trunk there are the Tukano, Dessana, Tuyuka, Wanana, Bará, Kubeu, Barassana, Piratapuia, Tariana, Miriti-Tapuya, Arapasso, Karapanã, Makuna and Siriano ethnicities. The Aruak trunk is composed of the Baré, Baniwa, Werekena, and Kuripako ethnicities. The Hüpd'ah, Nädeb, Yuhup'deh and Dãw ethnic groups belong to the Maku linguistic branch. Currently, this SIHD has 54 multidisciplinary indigenous health teams that are composed of doctors, nurses, nursing assistants, indigenous health agents, indigenous sanitation agents, dentists, oral health assistants, and laboratory technicians. In this SIHD, the participants were recruited in five health centers and in 16 indigenous villages.

**Upper Rio Solimões SIHD.**    This SIHD serves the second-largest indigenous population in Brazil, with a total of 70,891 indigenous people living in 241 villages with 13 health centers that are located in seven municipalities (Tabatinga, Benjamin Constant, São Paulo de Olivença, Amaturá, Santo Antônio do Içá, Tonantins, and Japurá), with seven indigenous ethnicities (Tikuna, Kokama, Kaixana, Kambeba, Kanamari, Witoto, and Maku-Yuhup). This population is distributed in 44 indigenous reserves that are located in the region of the

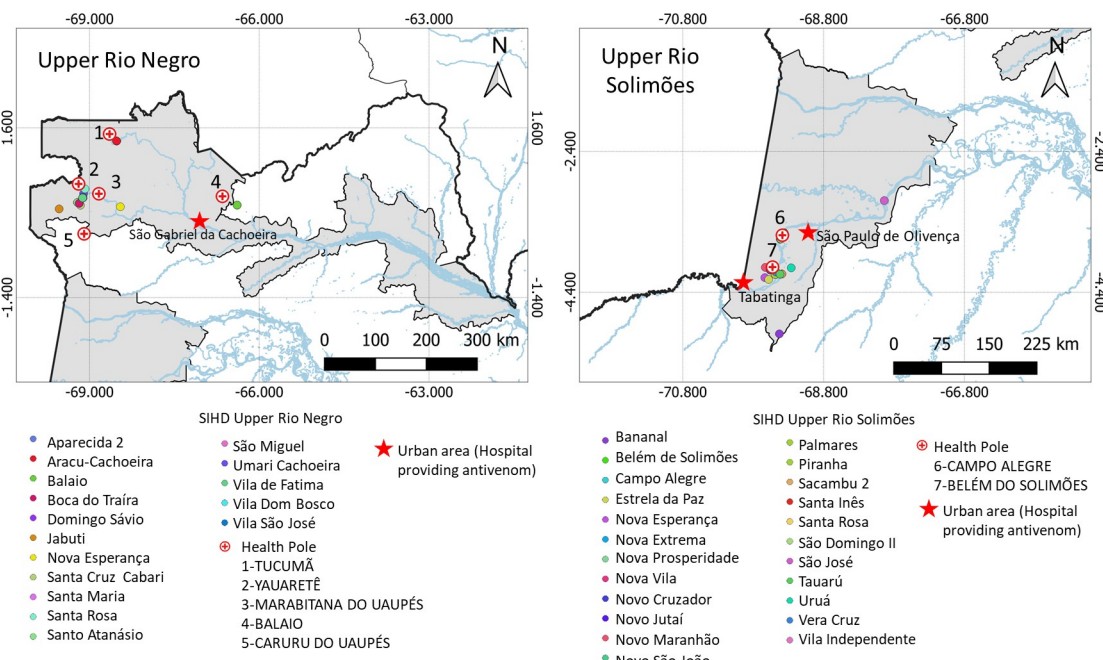

**Fig 2. Indigenous villages within the special indigenous health districts where the participants were recruited for the study.** A: Location of the five health centers and the 16 indigenous villages of the Upper Rio Negro SIHD, with participants recruited for the study; B: Location of the two health centers and the 23 indigenous villages of the Upper Rio Solimões SIHD, with participants recruited for the study. The geographical coordinates of each village were obtained during the visits using GPS (Garmin GPSMAP 64x). The base used to create the map was obtained from the Brazilian Institute of Geography and Statistics, which is freely accessible for creative use in shapefile format, in accordance with the Access to Information Law (12,527/2011) (https://www.ibge. gov.br/geociencias/downloads-geociencias.html?caminho=cartas_e_mapas/bases_cartograficas_continuas/bc250/versao2021/).

Jacurapá, Camatiã, Jandiatuba, Tacana, Igarapé de Belém, Uaiti Paranã, Jacapari, Matintin, Içá and Apoporis Rivers. Th health teams in this SIHD include doctors, nurses, dentists, psychologists, pharmacists, nutritionists, laboratory technicians, nursing technicians, oral health assistants, indigenous health agents, and indigenous sanitation agents. In this SIHD, the participants were recruited in two health centers and in 23 indigenous villages.

## Study design, participants and procedures

In this survey, the participants were identified from the SBE cases reported to the Indigenous Health Care Information System (*Sistema de Informação da Atenção à Saúde Indígena*; SIASI), cases identified by the indigenous health agents and by other health professionals working in the villages, and by using a snowball sampling technique–a non-probability sampling technique in which previously included participants in each village identified other eligible subjects from among their acquaintances [19]. Fig 2 shows the communities within the SDIH where the participants were recruited for the study.

Data were collected through semi-structured interviews by three trained nurses, via the application of a questionnaire with open and closed questions. Children and adolescents were interviewed in the presence of their parents or legal guardians, who assisted them in their responses. In the case of reported deaths, information regarding the deceased person was collected by interviewing the relatives. The questionnaire assessed information on demographic and socio-economic characteristics, including gender, age, ethnicity, marital status, literacy, occupation, religion, monthly income, municipality and community of residence, fluency in Portuguese, and characteristics of their houses. Participants were asked about the number of

SBEs they had suffered, the date of the most recent SBE, the geographic location where the bite occurred and the work he/she performed, the popular name of the perpetrating snake, use of protective measures, the anatomical region of the bite, local and systemic signs and symptoms, use of indigenous medicine, visit to a hospital for treatment, and the time elapsed from bite occurring to medical care being received. A board containing photographs of snakes was shown to the patient to assess whether they recognized the specimen responsible for the envenomations [20]. In cases in which the participant was not treated at hospital, the reason given for the non-utilization of the health service was also registered.

Whenever the participant was not fluent in Portuguese, an indigenous health agent participated as an interpreter.

These communities present a model of land occupation and natural-resource use that is predominantly subsistence-oriented. Main subsistence activities are fishing, agriculture (cassava, yam, corn, watermelon, sugarcane, banana, pineapple), the extraction of forest resources for basketry and crafts, hunting, and livestock raising (chickens, pigs and goats). During the visits, cassava plantations were observed in most communities, the production of which serves essentially to prepare cassava flour for their own consumption. Fishing and game hunting are important sources of food for these populations. Some villages carry out fishing and game hunting activities both for subsistence and for trade with non-indigenous communities. There are also açaí plantations, and fruit and vegetable gardens in most villages. The exchange of goods between villages is a common practice, including between different ethnic groups. The Baniwas are excellent artisans (of baskets, cassava graters, hammocks, and decorative objects) and cultivate several varieties of peppers, whose trade is commercially important for these villages. Some indigenous groups, especially the Tikunas, who interact more frequently with urban populations, either sell their products at markets in the cities (fruits such as ingá, mapati, pupunha, banana, umari, pineapple, araza, abiu, guava, soursop; cassava and cassava flour; handcrafts, and others) or buy industrialized products (rice, pasta, sugar, salt, soft drinks, frozen chicken, and others) for their own consumption and resale in the villages. They usually visit cities by waterway using small motorized boats. Fuels like gasoline are highly prized in villages. The houses are usually built of wood, and they are made on solid, floating or stilt bases. The Hüpd'ah build their houses out of mud. Many ethnic groups do not obey international borders and circulate in the Brazilian, Colombian and Peruvian territories. Only the most-populated villages have electricity and schools that provide basic education.

## Statistical analysis

Comparison of the participants' characteristics, history of SBEs, access to hospital treatments, and time elapsed from the bite occurring to medical assistance being received between the SIHDs was made using a Chi-square test (corrected by Fisher's exact test, if necessary). Analysis of association was performed to assess the factors associated to lack of '*access to healthcare*', defined here as the completion of the therapeutic itinerary from the moment of the bite to the participant's admission to the hospital. The analysis included the estimates of access to healthcare in the studied population and identification of factors associated to access. The crude odds ratios (ORs) with their respective 95% confidence interval (95%CI) were determined considering lack of access to health care as a dependent variable. Logistic regression was used for the multivariate analyses and the adjusted ORs with 95% CI were also calculated. All variables associated with the outcomes at a significance level of $p < 0.20$ in the univariate analysis were included in the multivariable analysis. Statistical significance was considered if $p < 0.05$ in the statistical tests. The analysis was performed using STATA software (StataCorp. 2013: Release 13. College Station, TX, USA).

## Results

The STROBE checklist is presented in the S1 Checklist.

### Characteristics of the participants

A total of 187 individuals with a previous history of SBEs were identified in the indigenous villages, with 131 (70.1%) living in the Upper Rio Solimões SIHD and 56 (29.9%) in the Upper Rio Negro SIHD. Participants were mostly male (61.0%), aged 40-59-years old (26.2%), and illiterate or with $\leq$ 4 years of schooling (42.8%). Tikuna was the most-represented ethnic group in the Upper Rio Solimões SIHD (92.4%). The Hüpd'ah (33.9%) and Tukano (21.4%) ethnic groups were the most-represented in the Upper Rio Negro SIHD. Most of them were involved in agriculture/fishing/hunting/forestry activities (56.1%), and married or in stable relationships (58.8%). Regarding religion, participants declared themselves as Catholics (42.8%), Protestants (40.6%), or of the Saint Cross Order (16.0%). Most participants received welfare benefits, pensions family allowance as their principal source of income (40.6%), followed by income from primary sector activities (38.0%). Wooden houses were the main type of housing (85.6%). A proportion of 67.9% of the participants understand Portuguese, and 65.8% is able to fluently speak Portuguese. The characteristics of the study participants are presented in Table 1.

### History of snakebites

A total of 15 (8.1%) participants reported $\geq$2 SBEs during their lifetime. Most of the SBEs that happened were reported 1–5 years before this survey (101; 62.0%). In the two regions, SBEs occurred more frequently in non-floodable areas (91; 48.7%), in which agricultural, hunting and forestry activities are performed. *Bothrops* SBEs predominated in the two regions (182; 97.3%), ranging from 96.2% in the villages of the upper Solimões River to 100% in the villages of the upper Negro River. SBEs were reported mostly in the lower limbs (159; 85.0%) (Table 2).

Chi-square test (corrected by Fisher's exact test, if necessary).

### Use of indigenous medicine

The use of traditional treatments was reported by 57.8% of the participants interviewed. In 39 (36.1%) cases, the participants did not reveal the identity of the plant-derived medicines. Plants used in these preparations were not identified because the participants did not remember the name, it was not possible to find a corresponding name in Portuguese, the remedy was prepared by another person who did not inform them of the composition or it is a remedy whose composition is confidential. Prayers and chants, usually combined with tobacco smoking, were used by 18 (16.7%) participants. The indigenous medicines used by the study participants are presented in Table 3.

Some plants were not identified because the participants did not remember the name, it was not possible to find a corresponding name in Portuguese, the remedy was prepared by another person who did not inform the composition or it is a remedy whose composition is confidential.

### Access to healthcare and associated factors

A total of 164 (87.7%) participants reported that they had access to healthcare and received assistance in a hospital in the urban area of the municipalities. The frequency was 95.4% in the Upper Rio Solimões SIHD, and 69.6% in the Upper Rio Negro SIHD ($p<0.0001$). Most of the

**Table 1. Characteristics of indigenous victims of snakebite envenomations in two special indigenous health districts, state of Amazonas, Western Brazilian Amazon.**

| Variable | Total | Upper Rio Solimões | Upper Rio Negro |
|---|---|---|---|
| | (n = 187; 100%) | (n = 131; 70.1%) | (n = 56; 29.9%) |
| **Gender** | | | |
| Male | 114 (61.0%) | 79 (60.3%) | 35 (62.5%) |
| **Age (years)** | | | |
| <18 | 32 (17.1%) | 23 (17.6%) | 9 (16.1%) |
| 18–29 | 43 (23.0%) | 32 (24.4%) | 11 (19.6%) |
| 30–39 | 24 (12.8%) | 15 (11.5%) | 9 (16.1%) |
| 40–59 | 49 (26.2%) | 40 (30.5%) | 9 (16.1%) |
| ≥60 | 39 (20.9%) | 21 (16.0%) | 18 (32.1%) |
| **Ethnic group** | | | |
| Tikuna | 121 (64.7%) | 121 (92.4%) | - |
| Kokama | 10 (5.3%) | 10 (7.6%) | - |
| Tukano | 12 (6.4%) | - | 12 (21.4%) |
| Piratapuya | 4 (2.1%) | - | 4 (7.1%) |
| Hüpd'ah | 19 (10.2%) | - | 19 (33.9%) |
| Tariano | 6 (3.2%) | - | 6 (10.7%) |
| Baniwa | 8 (4.3%) | - | 8 (14.3%) |
| Dessano | 3 (1.6%) | - | 3 (5.4%) |
| Wanano | 2 (1.1%) | - | 2 (3.6%) |
| Tuyuca | 2 (1.1%) | - | 2 (3.6%) |
| **Schooling (years)** | | | |
| Illiterate | 38 (20.3%) | 24 (18.3%) | 14 (25.0%) |
| 1–4 | 42 (22.4%) | 28 (21.3%) | 14 (25.0%) |
| 5–8 | 59 (31.6%) | 45 (34.4%) | 14 (25.0%) |
| >8 | 48 (25.7%) | 34 (26.0%) | 14 (25.0%) |
| **Occupation** | | | |
| Farmer/hunter | 105 (56.1%) | 76 (58.1%) | 29 (51.8%) |
| Fisher | 10 (5.3%) | 10 (7.6%) | 0 (0.0%) |
| Student | 39 (20.9%) | 25 (19.1%) | 14 (25.0%) |
| Retired | 30 (16.1%) | 18 (13.7%) | 12 (21.4%) |
| Professor | 3 (1.6%) | 2 (1.5%) | 1 (1.8%) |
| **Marital status** | | | |
| Single | 55 (29.5%) | 40 (30.6%) | 15 (26.8%) |
| Married | 110 (58.8%) | 76 (58.0%) | 34 (60.7%) |
| Widower | 10 (5.3%) | 7 (5.3%) | 3 (5.4%) |
| Not applicable | 12 (6.4%) | 8 (6.1%) | 4 (7.1%) |
| **Religion** | | | |
| Catholic | 80 (42.9%) | 32 (24.4%) | 48 (85.7%) |
| Saint Cross Order# | 30 (16.0%) | 30 (22.9%) | - |
| Protestant | 76 (40.6%) | 68 (51.9%) | 8 (14.3%) |
| Other | 1 (0.5%) | 1 (0.8%) | - |
| **Income source** | | | |
| Primary sector¥ | 71 (38.0%) | 65 (49.6%) | 6 (10.7%) |
| Services | 14 (7.5%) | 10 (7.6%) | 4 (7.1%) |
| Welfare, pensions and family allowance | 76 (40.6%) | 55 (42.0%) | 21 (37.6%) |
| Subsistence activities | 26 (13.9%) | 1 (0.8%) | 25 (44.6%) |

(*Continued*)

**Table 1.** (Continued)

| Variable | Total | Upper Rio Solimões | Upper Rio Negro |
|---|---|---|---|
| | (n = 187; 100%) | (n = 131; 70.1%) | (n = 56; 29.9%) |
| **Type of housing** | | | |
| Wood | 160 (85.6%) | 115 (87.8%) | 45 (80.4%) |
| Masonry | 17 (9.1%) | 13 (9.9%) | 4 (7.1%) |
| *Palafita* (on stilts) | 7 (3.7%) | 3 (2.3%) | 4 (7.1%) |
| Mud house | 3 (1.6%) | - | 3 (5.4%) |
| **Understands Portuguese** | 127 (67.9%) | 76 (58.0%) | 51 (91.1%) |
| **Speaks Portuguese** | 123 (65.8%) | 75 (57.3%) | 48 (85.7%) |

\# Saint Cross Order is a Messianic order that proliferates among indigenous communities in the Upper Rio Solimões. It was founded by a missionary in Minas Gerais, southeastern Brazil, in the early 70's, who, after traveling through several countries in South America, ended up settling in the Upper Solimões region.

¥ Fishing, agriculture, forestry, and hunting.

participants who had access to hospitals sought it in less than 6 hours (50.3%). However, the frequency of participants that sought hospital care more than 24 hours after the snakebite was significantly higher for those living in the Upper Rio Negro SIHD (21.7% versus 5.1%; $p = 0.001$) (Fig 3).

**Table 2. Characteristics of the 187 study participants according to their history of snakebites.**

| Variable | Total | Upper Rio Solimões HD | Upper Rio Negro HD | *p*-value |
|---|---|---|---|---|
| | (n = 187; 100%) | (n = 131; 70.1%) | (n = 56; 29.9%) | |
| **Number of snakebites** | | | | 0.34 |
| 1 | 172 (92.0%) | 123 (93.9%) | 49 (87.5%) | |
| 2 | 13 (6.9%) | 7 (5.3%) | 6 (10.7%) | |
| 3 | 2 (1.1%) | 1 (0.8%) | 1 (1.8%) | |
| **Date of the last snakebite** | | | | <0.001 |
| <3 months | 9 (5.5%) | 8 (6.6%) | 1 (2.4%) | |
| 3–6 months | 8 (4.9%) | 8 (6.6%) | - | |
| 6 months-1 year | 10 (6.1%) | 10 (8.3%) | - | |
| 1–5 years | 101 (62.1%) | 74 (61.2%) | 27 (64.2%) | |
| 6–10 years | 17 (10.4%) | 16 (13.2%) | 1 (2.4%) | |
| ≥10 years | 18 (11.0%) | 5 (4.1%) | 13 (31.0%) | |
| **Place where snakebite occurred** | | | | 0.014 |
| Non-floodable area | 91 (48.7%) | 55 (42.0%) | 36 (64.3%) | |
| Floodplain area | 41 (21.9%) | 36 (27.5%) | 5 (8.9%) | |
| Household area | 30 (16.0%) | 21 (16.0%) | 9 (16.1%) | |
| River bank | 25 (13.4%) | 19 (14.5%) | 6 (10.7%) | |
| **Type of snakebite** | | | | 0.53 |
| *Bothrops* | 182 (97.4%) | 126 (96.1%) | 56 (100.0%) | |
| *Lachesis* | 3 (1.6%) | 3 (2.3%) | - | |
| *Micrurus* | 1 (0.5%) | 1 (0.8%) | - | |
| Unknown | 1 (0.5%) | 1 (0.8%) | - | |
| **Anatomical region of the bite** | | | | 0.13 |
| Upper limbs | 24 (12.8%) | 21 (16.0%) | 3 (5.4%) | |
| Lower limbs | 159 (85.1%) | 107 (81.7%) | 52 (92.8%) | |
| Other | 4 (2.1%) | 3 (2.3%) | 1 (1.8%) | |

**Table 3. Indigenous medicines used by 108 study participants.**

| Treatment | Form of administration | Number (%) |
|---|---|---|
| Plant-derived preparations | Infusions for oral use, or application of plasters made from crushed plants at the bite site. | 39 (36.1) |
| Prayers and chants | Prayers and chants, usually combined with tobacco smoking, performed by the shaman or other healing agents. | 18 (16.7%) |
| "*Potato*" or "*little potato*" (wild plant from the Brazilian Amazon) | Infusions of the tuber for oral use, or application of plasters made from crushed plants at the bite site. | 17 (15.7) |
| Gasoline | Application at the bite site. | 13 (12.0) |
| Salt | Application at the bite site. | 12 (11.1) |
| Snake tissues | Application of skin, entrails, and feces at the bite site; ingestion of the snake's blood and heart. | 8 (7.4) |
| "*Sororoca*" or "*bananeira-brava*", a banana-like herb (*Phenakospermum guyannense*) | Infusions of the root for oral use. | 8 (7.4) |
| Salt water | Ingestion of the solution. | 5 (4.6) |
| Mastruz, Jesuit's tea, or Mexican-tea (*Dysphania ambrosioides*) | Application of plasters made from crushed leaves at the bite site. | 4 (3.7) |
| Açaí palm tree (*Euterpe oleracea*) | Infusions of the root for oral use. | 4 (3.7) |
| Tobacco (*Nicotiana tabacum*) | Application of plasters made from crushed leaves at the bite site, ingestion of tobacco macerated in water or smoking. | 4 (3.7) |
| Cathedral bells or corama (*Kalanchoe pinnata*) | Application of plasters made from crushed leaves at the bite site. | 3 (2.8) |
| Peach tomato or cubiu (*Solanum sessiflorum*) | Application of plasters made from crushed leaves at the bite site. | 3 (2.8) |
| Charcoal | Application of powdered charcoal at the bite site. | 3 (2.8) |
| Sorb tree (*Sorbus domestica*) | Application of the mixture of sorb tree latex with salt at the bite site. | 2 (1.9) |
| Pacori (*Pacouria boliviensis*) | Application of fruit pulp at the bite site. | 2 (1.9) |
| Ginger (*Zingiber officinale*) | Application of scrapings of the tubercle at the bite site. | 2 (1.9) |
| Capeba or pariparoba (*Piper umbellatum*) | Application of plasters made from crushed leaves at the bite site. | 1 (0.9) |
| Peccary (*Tayassu pecari*) | Use of the tooth in a glass of water (without ingestion of the tooth). | 1 (0.9) |
| Coffee (*Coffea* sp.) | Application of coffee powder at the bite site. | 1 (0.9) |
| Breu, almecega or almiscar (*Protium heptaphyllum*) | Application of the resin dissolved in gasoline at the bite site. | 1 (0.9) |
| Cayman (*Melanosuchus niger*) | Use of the tooth in the glass of water (without ingestion of the tooth). | 1 (0.9) |
| Gentian violet | Application of the solution at the bite site. | 1 (0.9) |
| Papaya (*Carica papaya*) | Application of the latex of the green fruit at the bite site. | 1 (0.9) |
| Nance, maricao cimun or murici (*Byrsonima crassifolia*) | Application of scrapings of the fruit at the bite site. | 1 (0.9) |

In the multivariate analysis, living in Upper Rio Negro SIHD was independently associated with the lack of access to healthcare [aOR 19.33 (95%CI = 4.11–90.85); *p*-value<0.01] (Table 4).

The main reason for not accessing healthcare as stated by the participants was the availability of indigenous medicine as the only choice in the village (75.0%), followed by lack of financial resources and means of transportation (28.1%), resistance to seeking medical assistance (6.3%), failure to recognize the situation as life-threatening (3.1%), and giving up on seeking medical help midway (3.1%). Thirteen participants cited more than one reason for not seeking medical attention.

## Deaths from snakebites

In the survey, four deaths from SBEs were informed, resulting in a case-fatality rate of 2.1% (4/187). In the Upper Rio Solimões SIHD, the estimated case-fatality rate was 1.5% (2/131) and, in the Upper Rio Negro SIHD, the rate was 3.6% (2/56). In summary, the patients' ages ranged from 29 to 89 years old. The adults were bitten during work activities. Only two of the individuals who died sought medical assistance at a hospital (Table 5).

## A) Access to hospital

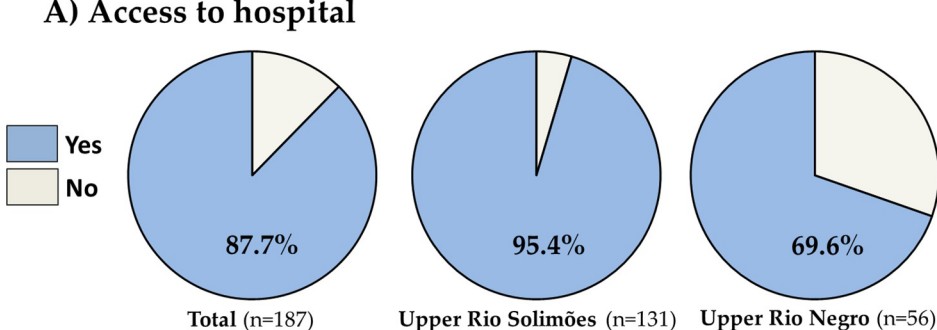

## B) Time from bite to hospital admission

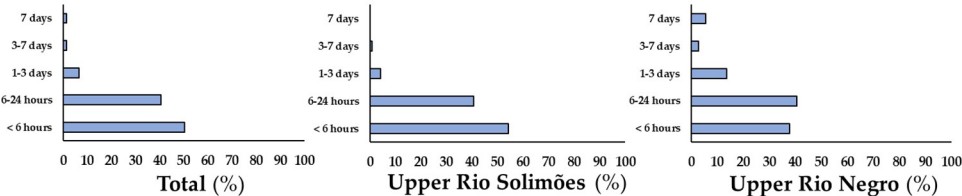

**Fig 3.** Access to healthcare (A) and time elapsed from the snakebite to hospital admission (in hours), according the special indigenous health district (B). The Chi-square test demonstrated a significantly higher frequency of access to hospital for participants living in the area of the Upper Rio Solimões SIHD compared to those of the Upper Rio Negro SIHD ($p<0.0001$), and the frequency of participants that had access to hospital with more than 24 hours after bite was significantly higher for those living in the Upper Rio Negro SIHD ($p = 0.001$).

## Discussion

### Snakebite demographics in the study area

In this study, we found that all age groups were affected by SBEs with a similar frequency. In general, previous studies have shown that there is a predominance of cases among adult men, which is usually explained by associating these cases with an occupational risk [17,21]. This difference in the age profile of cases among indigenous populations is probably due to the high exposure that children have in the villages. Since childhood, indigenous people have been taught by their parents to carry out agricultural, hunting and fishing activities, which puts them at a risk level that is similar to that of adults. Ethnographic studies demonstrate that indigenous children enjoy very high mobility in the villages, which allows them to go to different houses and be present in almost all moments of social life [22]. Additionally, previous works have demonstrated that indigenous people have a disproportionate incidence of SBEs in relation to non-indigenous people, which must also occur due to these activities carried out for subsistence, as well as the way of life in the villages, in close and continuous contact with the forest [3]. The leisure activities of indigenous children take place in the residential spaces, which are not places that are free from the presence of snakes. When bitten, children tend to have greater severity and complications due to their smaller body mass in comparison to adults since the same volume of venom is inoculated [23,24].

### Remarks on the indigenous medicine

Treatment of SBEs in indigenous villages is characterized by the use of tobacco smoking, chants and prayers, combined with animal tissues and bitter plants [25]. In this study, a

**Table 4. Factors associated to lack of access to healthcare in participants living in two special indigenous health districts, state of Amazonas, Western Brazilian Amazon.**

| Variable | OR | 95%CI | p-value | aOR | 95%CI | p-value |
|---|---|---|---|---|---|---|
| **Special Indigenous Health District** | | | | | | |
| Upper Rio Solimões | 1.00 | . | . | 1.00 | . | . |
| Upper Rio Negro | 9.08 | 3.35–24.63 | **<0.01** | 19.33 | 4.11–90.85 | **<0.01** |
| **Gender** | | | | | | |
| Female | 1.23 | 0.51–2.98 | 0.64 | | | |
| **Age (years)** | | | | | | |
| <18 | 1.00 | . | . | 1.00 | . | . |
| 18–29 | 0.73 | 0.14–3.85 | 0.71 | | | |
| 30–39 | 0.88 | 0.14–5.72 | 0.89 | | | |
| 40–59 | 1.35 | 0.31–5.83 | 0.69 | | | |
| ≥60 | 2.90 | 0.71–11.79 | 0.14 | 0.21 | 0.02–2.32 | 0.20 |
| **Schooling (years)** | | | | | | |
| Illiterate | 1.00 | . | . | 1.00 | . | . |
| 1–4 | 1.10 | 0.31–3.95 | 0.88 | | | |
| 5–8 | 0.89 | 0.26–3.03 | 0.85 | | | |
| >8 | 0.77 | 0.21–2.87 | 0.69 | | | |
| **Occupation** | | | | | | |
| Farmer | 0.37 | 0.15–0.92 | **0.03** | 0.20 | 0.03–1.52 | 0.12 |
| Fisher | 1.00 | . | . | 1.00 | . | . |
| Student | 0.78 | 0.25–2.43 | 0.66 | | | |
| Retired | 4.38 | 1.69–11.37 | **<0.01** | 5.85 | 0.36–95.46 | 0.22 |
| Teacher | 15.52 | 1.35–178.66 | **0.03** | 4.33 | 0.13–147.52 | 0.42 |
| Other | 0.70 | 0.09–5.74 | 0.74 | | | |
| **Marital status** | | | | | | |
| Single | 1.00 | . | . | 1.00 | . | . |
| Married | 4.84 | 1.08–21.79 | **0.04** | 5.46 | 0.54–55.66 | 0.15 |
| Widower | 6.63 | 0.81–53.9 | 0.08 | 7.97 | 0.52–123.04 | 0.14 |
| **Income source** | | | | | | |
| Services | 1.00 | . | . | 1.00 | . | . |
| Primary sector ¥ | 0.11 | 0.02–0.71 | **0.02** | 0.69 | 0.08–5.94 | 0.73 |
| Welfare, pensions and family allowance | 0.56 | 0.13–2.34 | 0.42 | . | . | . |
| Subsistence activities | 1.63 | 0.36–7.48 | 0.53 | . | . | . |
| **Number of snakebites** | | | | | | |
| 1 | 1.00 | . | . | . | . | . |
| 2 | 3.80 | 1.06–13.61 | **0.04** | 3.03 | 0.38–23.95 | 0.29 |
| 3 | 8.56 | 0.51–142.75 | 0.14 | 1.00 | . | . |
| **Date of the last snakebite** | | | | | | |
| <3 months | 1.00 | . | . | 1.00 | . | . |
| 3–6 months | 1.00 | . | . | | | |
| 6 months-1 year | 0.22 | 0.02–2.19 | 0.20 | 1.00 | . | . |
| 1–5 years | 0.20 | 0.06–0.65 | **0.01** | 0.38 | 0.09–1.58 | 0.18 |
| 6–10 years | 0.43 | 0.09–2.09 | 0.30 | | | |
| ≥10 years | 1.00 | . | . | | | |
| **Place where snakebite occurred** | | | | | | |
| Non-floodable area | 1.00 | . | . | 1.00 | . | . |
| Floodplain area | 0.26 | 0.06–1.19 | 0.08 | | | |

*(Continued)*

**Table 4.** (*Continued*)

| Variable | OR | 95%CI | *p*-value | aOR | 95%CI | *p*-value |
|---|---|---|---|---|---|---|
| Household area | 1.01 | 0.33–3.07 | 0.98 | | | |
| River bank | 0.21 | 0.03–1.68 | 0.14 | 0.40 | 0.03–4.59 | 0.46 |
| **Type of snakebite** | | | | | | |
| *Bothrops* | 1.00 | . | . | | | |
| *Lachesis* | 3.83 | 0.33–44.12 | 0.28 | | | |
| *Micrurus* | 1.00 | . | . | | | |
| **Anatomical region of the bite** | | | | | | |
| Upper limb | 1.00 | . | . | | | |
| Lower limb | 0.95 | 0.26–3.49 | 0.94 | | | |
| Other | 2.33 | 0.18–30.37 | 0.52 | | | |
| **Use of indigenous medicine** | 1.00 | . | . | | | |

¥ Fishing, agriculture, forestry and hunting.

**Table 5. Characteristics of the four deaths from snakebites as described by family members.**

| Death cases | Location | Description |
|---|---|---|
| 1 | Village of Nova Prosperidade, Campo Alegre Health Center, municipality of São Paulo de Olivença | Indigenous female of the Tikuna ethnicity, 58 years old, married, retired, illiterate, did not speak Portuguese, lived in a wooden house, was bitten on the leg by a pit viper (possibly *Botrops atrox*) while working in the cassava plantation in a lowland area. She felt pain, had swelling, did not seek health care and underwent treatment with indigenous medicine, ingesting the heart of a snake and a piece of the tail. She drank salt water and put salt and gasoline over the bite site. |
| 2 | Village of Nova Vila, Campo Alegre Health Center, municipality of São Paulo de Olivença | Indigenous female of the Tikuna ethnicity, 62 years old, widow, retired, illiterate, did not speak Portuguese, lived in a wooden house, was bitten on the foot by a pit viper (possibly *B. atrox*), in a peridomicile area while walking home. She felt pain, had swelling, and hemorrhage. She sought care at the Campo Alegre health center, and was referred to the urban area of the city for antivenom treatment only 8 hours after the bite. She did not undergo treatment with indigenous medicine. |
| 3 | Village of Aracu-Cachoeira, Tucumã Health Center, municipality of São Gabriel da Cachoeira | Indigenous male of Baniwa ethnicity, 29 years old, farmer, evangelical, spoke Portuguese, 8 years of schooling, single, lived in a wooden house, received family allowance, was bitten on the head by a pit viper (possibly *B. atrox*), on the river banks. He felt pain and had swelling. He was treated with indigenous medicine using roots of wild plants. It took 3 days to get to the urban area for antivenom treatment. |
| 4 | Village of Vila Dom Bosco, Yauaretê Health Center, municipality of São Gabriel da Cachoeira | Indigenous female of Baniwa Piratapuya ethnicity, 89 years old, married, without own income, Catholic, illiterate, spoke Portuguese, lived in a wooden house. She was bitten on the leg by a pit viper (possibly *B. atrox*) while on the farm. She felt pain and had swelling. She sought care at the Army Hospital in São Gabriel da Cachoeira within 24 hours. She did not undergo treatment with indigenous medicine. |

considerable variety of medicines, with a predominance of medicines derived from plants and animals, alone or in combination with chants and prayers, was used. Most of the plants used in the treatment of SBEs did not have their identity revealed to the interviewer. This could be a strategy to preserve its therapeutic arsenal or a limitation associated with the lack of a known translation of a popular name of that plant into Portuguese. Despite the more frequent contact of indigenous populations with the urban environment nowadays, the use of western medicines was not mentioned in this study. Indeed, there is a persistent role of indigenous healers in Amazonian societies after the introduction of western medicine and advanced stage of medical pluralism in villages that are located closer to urban areas [26]. In the case of SBEs, this persistence may be associated with the thought that this health problem can only be treated by indigenous therapy due to an underlying extraphysical cause; for instance, if it is possible to incorporate characteristics of the perpetrating snake by using parts of this animal as medicine, as commonly seen in this study, this will benefit the patient regarding SBE complications [25]. Moreover, there is a tension among different categories of healers and resistance to being treated by a biomedical health system that is not tailored to indigenous needs [25,27]. Our results provide evidence of the need to collaborate with indigenous healers to engage them in a healthcare model with a timely referral of SBE patients to a facility that is equipped with antivenom while maintaining the parallel offer of indigenous practices to patients.

An interesting finding was the self-care with the use of gasoline and table salt, which are obviously not originally components of the indigenous medicine. This appropriation of external elements deserves to be further investigated. With the intensification of contact with the non-indigenous population, it is possible that the indigenous people identify substances that in some way offer some relief, even if transient, from the symptoms of the SBE. Gasoline poured over the skin gains heat from it and evaporates, which ends up leaving a feeling of freshness in the area. On the other hand, the indigenous people could explain the mechanism of action of evaporating gasoline in the same way they understand the action of tobacco smoking, which 'sucks and removes' the venom from the patient's body [25]. One cannot discard the possibility that the use of gasoline can cause problems such as skin irritation, dryness, dermatitis, and even intoxication by inhalation.

## Unequal access to care for indigenous SBE patients in different health districts

In this study, lack of access to healthcare was significantly associated to living in the Upper Rio Negro health district compared to the Upper Rio Solimões health district. In most of the Brazilian territory, in which access to hospitals is possible by roads, SBE patients can be admitted directly to hospitals that have antivenom available and which belong to the emergency care network in the Unified Health System (*Rede de Atenção às Urgências no Sistema Único de Saúde)* [28]. In this medical context, transport of patients to the hospitals can be carried out by the patients' own means or by ambulances of the mobile emergency care service (*Serviço de Atendimento Móvel de Urgência*—SAMU), a public service for pre-hospital emergency transport. In the Amazon region, in a limited number of indigenous districts, there are ambulance service motorboats from the SAMU for indigenous healthcare (*Serviço de Atendimento Móvel de Urgência da Saúde Indígena*—SAMUSI) [29]. This is the case of the Upper Rio Solimões SIHD, which explains the better access to antivenom treatment in this district. In the Upper Rio Negro SIHD, health centers are located in areas so remote that transport is only possible by boats traveling enormous distances or by plane [27,30]. In addition, the indigenous peoples of the Upper Rio Solimões SIHD have more options for hospitals in the seven urban areas within its coverage area, while the Upper Rio Negro SIHD has only one point of care with

regular antivenom, which is in the municipality of São Gabriel da Cachoeira. Finally, the navigability of the Solimoes River and its tributaries is better than that of the Negro River basin, with narrow tributaries and several waterfalls along its course. Among the non-indigenous riverside populations living on the banks of the Solimões, Juruá and Purus Rivers, the proportion of SBE patients who did not have access to the health system (53%) was even higher than that observed in this study, which demonstrates the greater vulnerability of populations not assisted by a specific health subsystem, such as the indigenous peoples [21].

In addition to the lack of transport, barriers to medical care related to the sociocultural aspects of indigenous populations are also noted. For many populations, going to the city in search of treatment generates fear and doubt about the outcome of the case, since they need to leave their territories and stay away from their families for a long time. In severe SBE cases that evolve with complications, indigenous patients are referred from the municipalities of origin to medium and high complexity facilities in Manaus, the state capital [31]. For example, some dietary and behavioral interdictions, which are part of the therapeutic itinerary to prevent SBE complications, which include the prohibition of consumption of various fish and game animals and of contact with pregnant and menstruating women are deemed necessary [25]. In practice, respect for these prohibitions can only be guaranteed if the indigenous person is in his or her village, where this takes place in the daily lives of these peoples [25]. Health professionals report that, in SIHD health units, indigenous people are subject to conflicts related to these interdictions [27]. The resolution of these conflicts depends on the adapted structure of the health units, presence of interpreters and the training and sensitivity of the care team and managers to carry out the necessary mediations so that the treatment is carried out in accordance with their culture [32–34]. As an example, we can mention that the units do not always accommodate the indigenous people in hammocks to sleep, as required by some indigenous people, instead of conventional beds [33]. This generates uncertainties that affect the indigenous imaginary and that of their entire family when speculating about death far from the village, outside the home, since the life cycle of an indigenous person must end with his burial in his territory [35]. Thus, the possibility of death in a hospital can be understood as an important reason for resistance that affects the search for treatment in hospitals. Thus, it is to be expected that they use their ancestral knowledge as the only therapeutic resource, as observed in this study, such as self-care or in rituals and medicines prescribed by different classes of indigenous caregivers [25].

In this study, we adopted a cross-sectional approach with a non-probabilistic sampling based on the ease of access and existing contact within the population of interest. However, some indigenous villages were not included due to difficulties in terms of access, which prevented the study from having a greater number of participants. These more distant villages are precisely the ones that are most likely to have a greater number of bad cases and less favorable outcomes. As a result, our sampling method impairs the generalizability of our results. Moreover, unfortunately, the number of inhabitants per community was not obtained, which did not allow the calculation of the prevalence of snakebites in this population. The study also has a limitation in that it does not represent the medical reality of all indigenous ethnicities in the Brazilian Amazon, due to their cultural differences and possibilities of accessing the health system. Finally, the collection of information in this study depends on the memory of the participants, who in many cases were already elderly or had suffered their SBE more than a decade ago.

## Concluding remarks

In the two indigenous areas studied, there are records of SBE victims who did not receive medical attention, including child patients. Access to healthcare was better in the Upper Rio

Solimões district, in which i) a public service for pre-hospital emergency transport using motorboats is available; ii) there are more options for hospitals with regular antivenom supply in seven urban areas within its coverage area, compared to only one in the Upper Rio Negro SIHD; iii) the navigability of the Solimoes River and its tributaries is better than that of the Negro River basin. Sociocultural barriers to healthcare in both areas may include resistance to being transferred from their territories and having to stay away from their families. Nevertheless, the lack of an adapted structure of the health units, of interpreters and of training and sensitivity of the care team to guarantee dietary and behavioral interdictions that are required, and of an indigenous caregiver in the health units, still generate resistance on the part of indigenous people to seeking treatment in hospitals.

## Supporting information

**S1 Checklist. STROBE checklist for cross-sectional studies.**
(DOC)

**S1 Data. Study database used in the analysis.**
(XLSX)

## Acknowledgments

We are grateful for the support of the coordinators and professionals who work in the Upper Rio Solimões and Upper Rio Negro health districts, and of the presidents and counselors of these health districts. We would like to thank the technicians of the Health Research Coordination of the National Council for Scientific and Technological Development (COSAU/CNPq) and the National Indigenous Foundation (FUNAI) for providing the permits to carry out this study.

## Author Contributions

**Conceptualization:** Altair Seabra de Farias, Marcus Lacerda, Fan Hui Wen, Wuelton Monteiro, Jacqueline Sachett.

**Data curation:** Altair Seabra de Farias, Joseir Saturnino Cristino, Franciane Ribeiro Farias, André Sachett, Alexandre Vilhena Silva-Neto.

**Formal analysis:** Altair Seabra de Farias, Alexandre Vilhena Silva-Neto, Vinícius Azevedo Machado, Wuelton Monteiro.

**Funding acquisition:** Marcus Lacerda, Felipe Murta, Fan Hui Wen, Wuelton Monteiro, Jacqueline Sachett.

**Investigation:** Altair Seabra de Farias, Manoel Rodrigues Gomes Filho, Macio da Costa Arévalo, Joseir Saturnino Cristino, Fabíola Guimarães de Carvalho, Sediel Andrade Ambrosio, Fan Hui Wen, Wuelton Monteiro.

**Methodology:** Fan Hui Wen, Wuelton Monteiro, Jacqueline Sachett.

**Project administration:** Altair Seabra de Farias, Manoel Rodrigues Gomes Filho, Felipe Murta, Jacqueline Sachett.

**Resources:** Manoel Rodrigues Gomes Filho, André Sachett, Alexandre Vilhena Silva-Neto, Sediel Andrade Ambrosio.

**Software:** Franciane Ribeiro Farias, André Sachett, Alexandre Vilhena Silva-Neto, Fabíola Guimarães de Carvalho.

**Supervision:** Altair Seabra de Farias, Manoel Rodrigues Gomes Filho, Macio da Costa Arévalo, Sediel Andrade Ambrosio, Erica da Silva Carvalho, Felipe Murta, Wuelton Monteiro, Jacqueline Sachett.

**Validation:** Altair Seabra de Farias, André Sachett, Alexandre Vilhena Silva-Neto, Fabíola Guimarães de Carvalho, Wuelton Monteiro.

**Visualization:** Franciane Ribeiro Farias, Alexandre Vilhena Silva-Neto, Fabíola Guimarães de Carvalho, Erica da Silva Carvalho, Wuelton Monteiro.

**Writing – original draft:** Altair Seabra de Farias, Joseir Saturnino Cristino, André Sachett, Alexandre Vilhena Silva-Neto, Erica da Silva Carvalho, Vinícius Azevedo Machado, Wuelton Monteiro, Jacqueline Sachett.

**Writing – review & editing:** Marcus Lacerda, Felipe Murta, Vinícius Azevedo Machado, Fan Hui Wen, Wuelton Monteiro, Jacqueline Sachett.

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
