## [Decision Letter · Decision Letter 0]

13 Apr 2023

Dear Dr Monteiro,

Thank you very much for submitting your manuscript "Snakebite Envenomations and Access to Treatment in Communities of Two Indigenous Areas of the Western Brazilian Amazon: a Cross-Sectional Study" for consideration at PLOS Neglected Tropical Diseases. As with all papers reviewed by the journal, your manuscript was reviewed by members of the editorial board and by several independent reviewers. In light of the reviews (below this email), we would like to invite the resubmission of a significantly-revised version that takes into account the reviewers' comments. 

We cannot make any decision about publication until we have seen the revised manuscript and your response to the reviewers' comments. Your revised manuscript is also likely to be sent to reviewers for further evaluation.

Sincerely,

Abdulrazaq G. Habib

Academic Editor

José María Gutiérrez

Section Editor

Reviewer's Responses to Questions

**Key Review Criteria Required for Acceptance?**

**Methods**

-Are the objectives of the study clearly articulated with a clear testable hypothesis stated?

-Is the study design appropriate to address the stated objectives?

-Is the population clearly described and appropriate for the hypothesis being tested?

-Is the sample size sufficient to ensure adequate power to address the hypothesis being tested?

-Were correct statistical analysis used to support conclusions?

-Are there concerns about ethical or regulatory requirements being met?

Reviewer #1: - the study objectives could be made clearer

- the study design is appropriate for the stated objectives

- no clear explanation was given for the sample size

- there was statistical analysis to support conclusions 

- The ethical concerns were well addressed

Reviewer #2: No comments under the methods as the manuscript has well described the objectives and appropriate study design, selection of samples, defining population and appropriate analysis. No ethical issues have been noted since the authors have obtained the ethical approval.

Reviewer #3: • This study has a clear objective and reports relevant data on SBE on two indigenous populations. However, it is not clear how the estimation of underreporting was conducted from a statistical point of view. The characteristics of those who did not receive medical attention after the snakebite are clearly stated, but it is not clear if all of them are not-reported cases. I recommend including or changing the aim of the study to a more descriptive perspective on the management of SBE in two indigenous populations.

• The authors need to clarify if data collection was conducted using a survey or an interview (See lines 212 and 233). 

• Lines 256 to 282 provide a detailed description of some characteristics of these populations; however, I recommend including part of this information in the introduction section. Also, it can be used in the methods section to briefly describe variables such as type of housing, occupation, and the characteristics of the place where the snakebite occurred.

**Results**

-Does the analysis presented match the analysis plan?

-Are the results clearly and completely presented?

-Are the figures (Tables, Images) of sufficient quality for clarity?

Reviewer #1: - the analysis matched the analysis plan

- the results were clearly and completely presented

- However, the Tables will need some improvements

Reviewer #2: No specific comments results, analysis, presentation, figures and images.

Reviewer #3: • Tables 1 and 2 allow an adequate description of the population. Moreover, Table 3 describes relevant information on the use of indigenous medicine as a treatment by study participants. 

• I recommend including relevant data from Table 4 in the text.

• Pictures in Figure 3 can be reduced and present just those that illustrate the housing, transportation, and lifestyle characteristics of the study population. 

• The use of the abbreviations SBE and SBEs is confusing throughout the whole document. Please review. 

• Line 506: I assume the abbreviation AV refers to antivenoms, as it was not previously mentioned in the document.

• Line 339: is the age range 49-59 or 40-59 years old, as stated in Table 1?

**Conclusions**

-Are the conclusions supported by the data presented?

-Are the limitations of analysis clearly described?

-Do the authors discuss how these data can be helpful to advance our understanding of the topic under study?

-Is public health relevance addressed?

Reviewer #1: - conclusion supported the data presented

- there were no clearly stated limitations of the study

- the authors discussed the data appropriately 

- the public health relevance of the study outcome was not clearly stated

Reviewer #2: Yes agree. Conclusions were drawn from the analyzed results. Limitation of the study have been discussed adequately.

Reviewer #3: • Study limitations are not clearly stated in the document.

• A discussion on the generalizability and implications of results for the indigenous population is not presented. 

• The conclusion does not refer to the relevance of study findings to promote and recommend actions to access antivenoms soon after a snakebite occurs and to bring traditional medicine and Western medical treatment together. As stated in lines 484-487.

**Editorial and Data Presentation Modifications?**

Reviewer #1: - there will be a need for English language editing as the grammatical style of the authors made understanding their thought flow difficult.

Reviewer #2: None

Reviewer #3: (No Response)

**Summary and General Comments**

Reviewer #1: Comments

General comments: This is an important topic and is relevant in generating data on snakebite envenoming, given that hospital data alone does not tell the whole story of SBE globally. However, below are a few specific comments that should help strengthen the manuscript's quality. 

Specific comments:

Introduction: The recount of the existing literature on the global burden of SBE, policies to mitigate it, local Brazilian Amazonian literature on the burden, paucity of data on SBE, antivenom crises, and barriers to access to medical facility care regarding snakebite is adequate. However, I struggled to understand the specific objectives of the study from the stated study aim.

Aim – page 7, lines 132-134: The specific objectives of this study may need to be modified to improve their specificity, clarity, measurability and reproducibility. This way, the reader knows the information to expect in the results and discussion sections of the paper. Thus, I offer the following suggestions. 

(1) The statement "to estimate SBE underreporting" may need to be modified: Given that some of the participants in this study accessed orthodox care, which should have been documented and, as such, are reported cases, it will be inappropriate to describe them as under-reporting. I suggest rephrasing this objective, e.g. to read: "to estimate SBE reported by the community-dwellers….."

(2) For clarity, I suggest that you include that you are "assessing participants' health care seeking behaviour following a snakebite".

(3) Analyze barriers that prevent victims from obtaining healthcare in indigenous communities located in two indigenous health districts, in the Western Brazilian Amazon.

Methods: How did the authors arrive at a sample size of 187?

Results:

1. Table 1, schooling the second and third columns don't total 100%. Similarly, cross-check occupation, religion, etc. Please cross-check the data in all tables.

2. Table 4: the legend to this table is confusing and should be rephrased to reflect that it contains factors associated with lack of access to healthcare.

Discussion: 

(1) What were the limitations of this study?

(2) There were no clear statements about the policy implications of the study's outcome.

Reviewer #2: Include the total number of people included in this study 187 under the "Methods/principal findings" of the ABSTRACT.

Reviewer #3: This study details characteristics and information on how SBE is treated by two indigenous communities, as well as different factors that can facilitate or difficult access to treatment. However, the objective of this study is just a part of the main focus of the discussion observed throughout the document. 

In addition, I suggest analyzing study limitations and the possibility of generalized results to these indigenous communities. 

A final word of advice, since I had trouble following some sentences, I recommend a detailed review of the manuscript for grammar consistency.

PLOS authors have the option to publish the peer review history of their article (what does this mean?). If published, this will include your full peer review and any attached files.

Reviewer #1: Yes: Godpower Chinedu Michael

Reviewer #2: Yes: Kalana Maduwage

Reviewer #3: No
---

## [Decision Letter · Decision Letter 1]

17 Jun 2023

Dear Dr Monteiro,

Thank you very much for submitting your manuscript "Snakebite Envenomations and Access to Treatment in Communities of Two Indigenous Areas of the Western Brazilian Amazon: a Cross-Sectional Study" for consideration at PLOS Neglected Tropical Diseases. As with all papers reviewed by the journal, your manuscript was reviewed by members of the editorial board and by several independent reviewers. In light of the reviews (below this email), we would like to invite the resubmission of a significantly-revised version that takes into account the reviewers' comments. 

We cannot make any decision about publication until we have seen the revised manuscript and your response to the reviewers' comments. Your revised manuscript is also likely to be sent to reviewers for further evaluation.

Sincerely,

Abdulrazaq G. Habib

Academic Editor

José María Gutiérrez

Section Editor

Reviewer's Responses to Questions

**Key Review Criteria Required for Acceptance?**

**Methods**

-Are the objectives of the study clearly articulated with a clear testable hypothesis stated?

-Is the study design appropriate to address the stated objectives?

-Is the population clearly described and appropriate for the hypothesis being tested?

-Is the sample size sufficient to ensure adequate power to address the hypothesis being tested?

-Were correct statistical analysis used to support conclusions?

-Are there concerns about ethical or regulatory requirements being met?

Reviewer #1: How did the authors estimate the sample size?

Reviewer #2: -Are the objectives of the study clearly articulated with a clear testable hypothesis stated? Yes

-Is the study design appropriate to address the stated objectives? Yes

-Is the population clearly described and appropriate for the hypothesis being tested? Yes

-Is the sample size sufficient to ensure adequate power to address the hypothesis being tested? Yes

-Were correct statistical analysis used to support conclusions? Yes

-Are there concerns about ethical or regulatory requirements being met? Yes

Reviewer #3: The population is clearly stated and described; for me, the objective of this study is to describe the lack of access to healthcare in these two populations, as stated in lines 287-292, and not an estimation of underreporting.

**Results**

-Does the analysis presented match the analysis plan?

-Are the results clearly and completely presented?

-Are the figures (Tables, Images) of sufficient quality for clarity?

Reviewer #1: The issues highlighted in my comments about the Tables were not addressed

Reviewer #2: -Does the analysis presented match the analysis plan? Yes

-Are the results clearly and completely presented? Yes

-Are the figures (Tables, Images) of sufficient quality for clarity? Yes

Reviewer #3: - Line 338: SBEs were reported or happened?

- Review lines 353 and 354

- Review the words "high" in line 430, and "even though" in line 434.

- Some ideas seem to be incomplete: lines 438-440, 454-456, 501-503

- Lines 508-515: review how these very important ideas correspond to the paragraph. 

- Line 478: I assume the abbreviation AV refers to antivenoms, as it was not previously mentioned in the document.

- Throughout the document review the use of "peoples"; people is a plural form and does not requiere the "s". I suggest using population instead in most cases.

**Conclusions**

-Are the conclusions supported by the data presented?

-Are the limitations of analysis clearly described?

-Do the authors discuss how these data can be helpful to advance our understanding of the topic under study?

-Is public health relevance addressed?

Reviewer #1: (No Response)

Reviewer #2: -Are the conclusions supported by the data presented? Yes

-Are the limitations of analysis clearly described? Yes

-Do the authors discuss how these data can be helpful to advance our understanding of the topic under study? Yes

-Is public health relevance addressed? Yes

Reviewer #3: - Study limitations are still not clearly stated in the document.

- A discussion on the generalizability and implications of results for the indigenous population is not presented.

- Lines 543-547: This conclusion does not seem to come from the study design.

**Editorial and Data Presentation Modifications?**

Reviewer #1: (No Response)

Reviewer #2: I agree with all the modifications have been done to the revised version of the manuscript and I would recommend to accept this after the editorial decision.

Reviewer #3: (No Response)

**Summary and General Comments**

Reviewer #1: 1) I am not sure in PLos journals, authors are not allowed to highlight corrections of queries from reviewers in their manuscript.

2) After painstakingly, going through the revised manuscript, i found that the authors did not respond to the comments regarding the issues I raised during the first submission.

3) Without any response from the authors I am constrained to recommend a revision

Reviewer #2: I agree with all the modifications have been done to the revised version of the manuscript and I would recommend to accept this after the editorial decision.

Reviewer #3: I suggest reviewing the objective as I still consider the aim was to present data on access to healthcare more than identifying underreporting. In addition, I suggest analyzing study limitations.

Grammar consistency is better in this reviewed version, however, I still had trouble following some

sentences.

PLOS authors have the option to publish the peer review history of their article (what does this mean?). If published, this will include your full peer review and any attached files.

Reviewer #1: Yes: Godpower Michael

Reviewer #2: Yes: Kalana Maduwage

Reviewer #3: No
---

## [Decision Letter · Decision Letter 2]

27 Jun 2023

Dear Dr Monteiro,

We are pleased to inform you that your manuscript 'Snakebite Envenomations and Access to Treatment in Communities of Two Indigenous Areas of the Western Brazilian Amazon: a Cross-Sectional Study' has been provisionally accepted for publication in PLOS Neglected Tropical Diseases.

Best regards,

Abdulrazaq G. Habib

Academic Editor

José María Gutiérrez

Section Editor

Reviewer's Responses to Questions

**Key Review Criteria Required for Acceptance?**

**Methods**

-Are the objectives of the study clearly articulated with a clear testable hypothesis stated?

-Is the study design appropriate to address the stated objectives?

-Is the population clearly described and appropriate for the hypothesis being tested?

-Is the sample size sufficient to ensure adequate power to address the hypothesis being tested?

-Were correct statistical analysis used to support conclusions?

-Are there concerns about ethical or regulatory requirements being met?

Reviewer #1: (No Response)

**Results**

-Does the analysis presented match the analysis plan?

-Are the results clearly and completely presented?

-Are the figures (Tables, Images) of sufficient quality for clarity?

Reviewer #1: (No Response)

**Conclusions**

-Are the conclusions supported by the data presented?

-Are the limitations of analysis clearly described?

-Do the authors discuss how these data can be helpful to advance our understanding of the topic under study?

-Is public health relevance addressed?

Reviewer #1: (No Response)

**Editorial and Data Presentation Modifications?**

Reviewer #1: (No Response)

**Summary and General Comments**

Reviewer #1: (No Response)

PLOS authors have the option to publish the peer review history of their article (what does this mean?). If published, this will include your full peer review and any attached files.

Reviewer #1: **Yes: **Godpower Chinedu Michael

---

## [Editor Report · Acceptance letter]

8 Jul 2023

Dear Dr. Monteiro,

We are delighted to inform you that your manuscript, "Snakebite Envenomations and Access to Treatment in Communities of Two Indigenous Areas of the Western Brazilian Amazon: a Cross-Sectional Study," has been formally accepted for publication in PLOS Neglected Tropical Diseases.

Best regards,

Shaden Kamhawi

co-Editor-in-Chief

Paul Brindley

co-Editor-in-Chief
